# MultiHateClip: A Multilingual Benchmark Dataset for Hateful Video Detection on YouTube and Bilibili

## ABSTRACT

Hate speech is a pressing issue in modern society, with significant repercussions both online and offline. Recent research in hate speech detection has primarily centered on text-based media, largely overlooking multimodal content such as videos. Existing studies on hateful video datasets have predominantly focused on English content within a Western context and have been limited to binary labels (hateful or non-hateful), lacking detailed contextual information. This study presents MultiHateClip, an novel multilingual dataset curated through hate lexicons and human annotation. It aims to enhance the detection of hateful videos on platforms such as YouTube and Bilibili, encompassing content in both English and Chinese languages. Comprising 2,000 videos annotated for hatefulness, offensiveness, and normalcy, this dataset provides a cross-cultural perspective on gender-based hate speech. Through a detailed examination of human annotation results, we discuss the differences between Chinese and English hateful videos and underscore the importance of different modalities in hateful and offensive video analysis. Evaluations of state-of-the-art video classification models, such as *VLM* and *GPT-4V*, on MultiHateClip highlight the existing challenges in accurately distinguishing between hateful and offensive content and the urgent need for models that are both multimodally and culturally nuanced. MultiHateClip serves as a foundational step towards developing more effective hateful video detection solutions, emphasizing the importance of a multimodal and culturally sensitive approach in the ongoing fight against online hate speech.

**Disclaimer: This paper contains sensitive content that may be disturbing to some readers.**

## CCS CONCEPTS

• **Computing methodologies** → **Computer vision**; **Natural language processing**.

## KEYWORDS

video, multimodal, multilingual, hateful video detection

**ACM Reference Format:**
Anonymous Authors. 2024. MultiHateClip: A Multilingual Benchmark Dataset for Hateful Video Detection on YouTube and Bilibili. In *Proceedings of the 32th ACM International Conference on Multimedia (MM '24), October 28 - November 1, 2024, Melbourne, Australia.* ACM, New York, NY, USA, 9 pages. https://doi.org/XXXXXXX.XXXXXXX

## 1 INTRODUCTION

The rapid expansion of social media has revolutionized the way information is shared, enhancing connectivity among users within both offline and online communities. However, these platforms have increasingly become conduits for the dissemination of hateful content that targets individuals or groups based on race, religion, gender, and other characteristics [17, 31]. The proliferation of online hate speech not only fosters discord among communities but also escalates to real-world violent hate crimes, underscoring the urgent need to identify and mitigate such content.

Current research on detecting hateful content has primarily concentrated on text-based analysis [17, 31], with recent advancements extending to multimodal forms, such as memes [8, 9, 21, 28, 32]. However, the field of hateful video detection remains underexplored, largely due to the lack of comprehensive datasets. Videos harness the synergistic potential of visual, auditory, and textual components to spread hate speech and offensive content, subsequently piquing the curiosity of select researchers [4, 11, 40]. For instance. in a recent study, Das et al. [11] constructed an English video dataset to facilitate hateful video classification. However, the existing studies have largely focused on English videos based on Western context, and these datasets could only facilitate simple coarse-grained binary classification (hateful or non-hateful) without diving into the fine-grained analysis of the hate speech context, e.g., identifying targets or victims in the hate speech.

To fill these gaps, our study introduces MultiHateClip, a multilingual short clip video dataset that facilitates a more nuanced and comprehensive exploration of multimodal hateful video content. This dataset compiled English short clips from YouTube, a global platform known for its vast user-generated content and diverse viewership, and Chinese short clips from Bilibili, a leading Chinese video-sharing website that caters to a younger demographic with a focus on animation, comics, and games (ACG) content. In this study, we specifically focus on gender-based hate speech in the Western and Chinese cultural contexts, presenting an unprecedented cross-cultural perspective on hate speech in digital media. MultiHateClip contains 2,000 short clip English and Chinese videos, which not only enriches the understanding of hate speech's multifaceted nature but also marks the first initiative to construct a cross-cultural video dataset dedicated to examining hate speech, specifically targeting gender-related issues across Western and Chinese domains.

When constructing MultiHateClip, we utilize 60 hate-specific lexicons for each language to identify relevant videos on YouTube and Bilibili. Our annotation process involved a team of native-speaker annotators who were familiar with Western and Chinese popular cultures. Following [12]; we differentiate hate speech from offensive language and task the annotators to categorize the videos into three distinct groups: *hateful*, *offensive*, or *normal*. Videos identified as

*hateful* or *offensive* were requested to undergo further annotation to pinpoint segments with hateful content, ascertain the target of hate speech—such as *Woman*, *Man*, *LGBTQ+*, or others—and determine the contributing modalities—whether *visual*, *auditory*, or *textual*.

We conducted a thorough analysis on the MultiHateClip dataset and outlined key insights that could influence hateful video detection. We noted a surprisingly low frequency of hate speech videos on both YouTube and Bilibili, even after a comprehensive review of over 10,000 videos. Ultimately, we curated annotations for 1001 videos per language, but only around 300 of them were labeled as hateful or offensive. The analysis of the annotated dataset revealed a consistent pattern: a disproportionate amount of gender-based hate speech targeting women, echoing the wider societal issues of misogyny and gender discrimination. Further evaluation of modality contributions in these videos underscored the complexity of hate speech communication. For instance, 80% of the hateful/offensive Chinese videos on Bilibili combined multiple modalities, such as visual, auditory, and textual modalities, to convey their messages. This multimodal nature of hate speech highlights the necessity of a comprehensive approach that integrates multiple modalities to provide a deeper, more nuanced understanding of hate speech dynamics.

To assess the effectiveness of current models in hate video classification, we tested several state-of-the-art models on MultiHateClip. The *VLM* [41] and *GPT-4-V* [3] models performed best in distinguishing between merged hateful/offensive content and normal content for English and Chinese videos, respectively. The *VLM* model achieved a macro F1-score of 0.583 for English, while the *GPT-4V* attained a score of 0.567 for Chinese content. These results reveal critical limitations in contemporary classification approaches, specifically: the challenge of differentiating between hateful and offensive content; the inadequacy of pre-trained models on non-Western cultural data; and the inefficacy of late fusion in integrating multimodal data. These findings highlight the need for advancements in video content classification models to better address these identified weaknesses.

Our research contributions summarized as follows:

(1) We have constructed MultiHateClip, a multilingual dataset of hateful short clip videos. This dataset is enriched with detailed annotations for videos deemed hateful or offensive, detailing the specific segments with hate speech, the targeted victims, and the modalities that contribute to the content's offensiveness. Such comprehensive annotations are designed to serve as a foundational resource for subsequent research.

(2) Our exhaustive examination of MultiHateClip has unveiled multilingual and cultural-specific characteristics and the critical role of multimodal inputs in hate speech detection. These insights are instrumental in refining approaches to hateful video detection, offering guidance for the development of more effective hateful video detection models.

(3) We have critically evaluated existing video classification models, identifying key areas of weakness: the challenge in differentiating between hateful and offensive content, the limitations of pre-trained models concerning non-Western

cultural data, and the shortcomings of late fusion of multimodal representations. These evaluations not only underscore existing gaps but also chart potential avenues for future research to enhance hateful video detection methodologies.

## 2 RELATED WORK

**Text-based Hate Speech Detection.** Extensive research has focused on detecting hate speech within textual content, producing a variety of datasets from platforms like Twitter [43], Stormfront [13], and Fox News [20]. While many studies engage in binary classification (hate speech or not), works such as [12] and [18] attempt to distinguish between hateful, offensive, and normal speech. Other research targets specific forms of hate speech, such as misogyny [16, 39] and racism [38]. Warner and Hirschberg[37] expand on this by categorizing hate speech into seven victim categories. However, these efforts primarily focus on text, leaving a gap in datasets and analysis for video content[10].

**Video-based Hate Speech Detection.** In contrast to the abundant textual hate speech datasets, video-based datasets remain underdeveloped. For instance, [4] and [40] introduce datasets comprising 400 Portuguese and 300 English YouTube videos, respectively. However, their limited sizes constrain the training of robust multimodal classification models. These studies primarily rely on textual analysis for classification. [11] expands the field by compiling 1,083 BitChute videos into a dataset, although it focuses on English videos with binary hate classification, missing information like target victim identification. Refer to Table 1 for a summary of relevant datasets, incorporating ours (MultiHateClip), in the realm of hateful video detection.

**Multimodal Model Fusion in Hate Speech Detection.** Traditional fusion techniques in hate speech detection often involve concatenating representation vectors from pre-trained unimodal models into a composite model, a method known as *late fusion*. Recent studies have demonstrated efficacy in hateful video classification tasks [11]. Despite its effectiveness, emerging evidence highlights the potential of early fusion could also improve performance [19]. Vision-Language (VL) models, such as VideoBERT [33], ClipBERT [24], VLM [41], and UniVL [26], show promise in video analysis tasks, with VLM and GPT-4[3] standing out for their capabilities in understanding and classifying video content.

Addressing the gaps in video-based hate speech research, we introduce a dataset of 2,000 videos annotated for levels of hatefulness, offensiveness, and normalcy, spanning English and Chinese cultural contexts. This dataset not only includes video labels but also detailed annotations on the hateful/offensive segments, target victim and contributing modalities. Our upcoming experiments will leverage both late and early fusion techniques, applying VL models like VLM and GPT-4, to evaluate their efficacy in classifying video content accurately.

## 3 MULTIHATECLIP DATASET

### 3.1 Data Collection

**Data Sources**: We collected videos from the social media platforms YouTube and Bilibili. Launched in 2005, YouTube has emerged as the leading global video-sharing platform, boasting 2.7 billion monthly

**Table 1: Summary of datasets in hateful video detection. H:hateful, O:offensive, N:normal/non-hateful/non-offensive.**

| Work | Language | Size | Label | Hateful Segment | Targeted Victim | Contributing modality |
|------|----------|------|-------|-----------------|-----------------|-----------------------|
| OffVidPT [4] | Portuguese | 400 | O, N | × | × | × |
| Hate_speech_ dataset_videos [40] | English | 300 | H, N | × | ✓ | × |
| HateMM [11] | English | 1083 | H, N | ✓ | × | × |
| MultiHateClip | Chinese, English | 2002 | H, O, N | ✓ | ✓ | ✓ |

active users [2], and is known for its comprehensive content moderation policies [34]. Bilibili, established in 2010, has become a central hub for the Chinese animation, comics, and gaming community, with over 3.36 billion monthly active users [1], offering a unique insight into Chinese digital culture.

**Video Collection**: To source videos, we curated 60 pairs of gender-based hate lexicons based on [7] and [23], targeting synonymous terms across English and Chinese, such as 'mistress' and '情妇'. Utilizing the YouTube and Bilibili APIs, we conducted keyword searches using these lexicons, specifically targeting short clip videos no longer than 60 seconds, aligning with our focus on brief, potentially virulent content. This approach resulted in the collection of 5,500 English and 5,000 Chinese videos. Acknowledging the platforms' content moderation policies, which limit the presence of hate speech, we employed ChatGPT-3.5[1] to conduct an initial categorization based on video titles and transcripts. This step aimed to sift through the amassed videos, singling out those potentially featuring hateful or offensive content for closer examination. Ultimately, 2,000 videos from each language were selected for detailed manual annotation, ensuring a comprehensive analysis of hate speech trends within the dataset.

## 3.2 Human Annotation

**Annotation Guidelines**. The annotation process we have developed requires annotators to address four principal questions about each short clip video, aimed at comprehensively assessing and categorizing its content.

**Q1) Video Category Labeling**: Annotators are asked to classify each video into one of the following four categories based on its content: *Hateful*, *Offensive*, *Normal*, or *Normal-Counter-Narrative*. The category definitions are provided to guide the annotators:

- *Hateful*: Videos that incite discrimination or demean individuals or groups based on attributes such as race, ethnicity, nationality, religion, disability, age, veteran status, sexual orientation, gender identity, etc.
- *Offensive*: Videos that may cause discomfort or distress, yet do not qualify as hateful under the criteria defined above.
- *Normal*: Content devoid of hatefulness or offensiveness.
  - *Counter-Narrative*: Includes videos that, despite containing potentially hateful or offensive segments, primarily aim to counteract such occurrences without intending harm.

**Q2) Identification of Hateful/Offensive Segment**: For videos classified as hateful or offensive, annotators are tasked with determining the precise segment containing such content. They must specify the start and end times of the segment where the hateful or offensive statements occur. This requirement ensures a targeted

analysis of the content, facilitating a more detailed examination of the nature and context of hate speech within the video.

**Q3) Identification of the Target Victim**: For videos identified as hateful or offensive, annotators are required to determine the target of the content. Given the dataset's focus on gender-related hate speech, annotators should specify the intended victim group—*Man*, *Woman*, or *LGBTQ+*. Additionally, there is an option to identify other groups if the video targets individuals or communities outside these categories. This step ensures a nuanced understanding of hate speech targeting, allowing for a comprehensive analysis of the videos' impact on various demographics.

**Q4) Determination of Contributing Modality**: In this part of the annotation process, annotators are asked to identify which modality—or modalities—of the video contribute to its hateful or offensive nature. They will classify the contribution as stemming from one or more of the following three categories: *text* (including titles and transcripts), *visual* content, or *audio* content. This step is crucial for understanding how hate speech is conveyed through different channels within a video, whether through written words, visual imagery, or spoken language, offering insights into the multimodal dynamics of hate speech dissemination.

**Annotators Recruitment and Training**. Two PhD students, proficient in hate speech, served as expert annotators, supported by a team of 12 undergraduate students. The undergraduate team was balanced in terms of language proficiency—split evenly between the two languages of the study—and gender representation within each language group.

Before starting the actual annotation, annotators participated in a training session to familiarize themselves with the annotation guidelines and procedures. They were then tasked with annotating a test set of 20 videos to gauge their understanding and application of the guidelines. Expert annotators reviewed the test annotations to ensure accuracy and consistency across the team.

**Annotators Process**. The annotation process was designed for robustness: each video initially received annotations from two different annotators. In instances of disagreement regarding the categorization (hateful, offensive, or normal), a third annotator was enlisted to provide an additional perspective. If disagreements persist, the matter is escalated to the expert annotators for final annotation. The ultimate categorization of each video was determined through a majority vote, ensuring a high level of consensus and reliability in the annotated dataset.

To manage the workload efficiently, we implemented a batch annotation strategy, allocating approximately 30 videos to each annotator daily. This task was designed to be manageable within a 30 to 40-minute commitment per day. Expert annotators played a crucial role in quality control, examining the day's annotations to verify label distribution and conducting random checks on selected video annotations. This daily review process allowed for immediate

---

[1] https://api.openai.com/v1/chat/completions

**Table 2: Statistics of English YouTube videos. H:Hateful, O:Offensive, N:Normal, CN:Counter Narrative, T:Total.**

|                        | H     | O     | N     | CN     | T     |
|------------------------|-------|-------|-------|--------|-------|
| Count                  | 77    | 230   | 687   | 7      | 1001  |
| Avg. Title len         | 8.17  | 8.3   | 8.81  | 11.43  | 8.66  |
| Avg. Transcript len    | 82.4  | 68.71 | 84.96 | 127.14 | 81.32 |
| Avg. Video len(sec)    | 35.7  | 31.97 | 35.13 | 46.14  | 34.53 |
| Avg. H/O segment(sec)  | 23.68 | 21.82 | -     | -      | 22.42 |

**Table 3: Statistics of Chinese Bilibili videos. H:Hateful, O:Offensive, N:Normal, CN:Counter Narrative, T:Total.**

|                        | H     | O     | N     | CN     | T     |
|------------------------|-------|-------|-------|--------|-------|
| Count                  | 108   | 170   | 710   | 13     | 1001  |
| Avg. Title len         | 13.38 | 14.19 | 17.35 | 19.46  | 16.41 |
| Avg. Transcript len    | 36.19 | 37.75 | 74.32 | 127.92 | 64.69 |
| Avg. Video len(sec)    | 25.15 | 29.32 | 33.24 | 36.15  | 31.74 |
| Avg. H/O segment(sec)  | 21.03 | 25.28 | -     | -      | 23.47 |

**Table 4: Distributions of videos targeting each victim group. H:Hateful, O:Offensive.**

|                | English | | Chinese | |
|----------------|---------|------|---------|------|
| **Target Victim** | **H** | **O** | **H** | **O** |
| Woman          | 39      | 108  | 50      | 83   |
| Man            | 28      | 55   | 46      | 40   |
| LGBTQ          | 30      | 26   | 29      | 25   |
| Others         | 22      | 26   | 36      | 7    |

feedback and rectification of any misconceptions, with particular attention paid to outliers or unexpected labeling trends. Annotators were periodically engaged in discussions to resolve any ambiguities or misinterpretations, ensuring a uniform understanding and application of the annotation guidelines.

## 3.3 Data Statistics and Analysis

**Inter-Annotator Agreement**: We assessed the reliability of our annotations using Cohen's kappa. For the initial annotations, before consolidating "*hateful*" and "*offensive*" categories, the kappa scores were 0.518 for English and 0.440 for Chinese in the multiclass classification system. After simplification into a binary system (combining "*hateful*" and "*offensive*"), the scores improved to 0.598 for English and 0.635 for Chinese. These results underscore the annotations' reliability and the effectiveness of our consensus-building process.

**Dataset Statistics**: Each video's transcript was obtained via Google Cloud Speech-to-Text[2]. We then documented key metrics such as the number of videos, word counts in titles and transcripts, video lengths, and lengths of segments identified as hateful or offensive. These statistics are presented in Table 2 for English videos and Table 3 for Chinese videos.

**Label Distribution**: MultiHateClip has a 3:7 ratio of *hateful/offensive* to *normal* videos, reflecting the platforms' strict content moderation policies. *Counter Narrative* videos constituted a minor fraction and were collapse under "*Normal*" for further analysis. Notably, "*Normal*" videos tend to have longer titles and transcripts, especially in Chinese, where the transcript length of "*Normal*" videos are nearly double that of "*Hateful*" or "*Offensive*" videos.

**Victim Group Analysis**: The distribution of hateful and offensive content targeting specific victim groups is detailed in Table 4.

[2]https://cloud.google.com/speech-to-text

**Table 5: Modality distribution of hateful and offensive videos. H:Hateful, O:Offensive.**

|                     | English | | Chinese | |
|---------------------|---------|-----|---------|-----|
| **Modalities**      | **H**   | **O** | **H** | **O** |
| Text                | 12      | 99  | 16      | 36  |
| Audio               | 1       | 2   | 0       | 0   |
| Vision              | 0       | 11  | 0       | 5   |
| Text ⊙ Vision       | 8       | 41  | 36      | 56  |
| Audio ⊙ Vision      | 0       | 2   | 0       | 0   |
| Text ⊙ Audio        | 24      | 31  | 15      | 23  |
| Text ⊙ Audio ⊙ Vision | 32    | 44  | 42      | 50  |

**Table 6: Top 10 highest tf-idf score keywords in videos' transcripts and titles. with Hateful keywords highlight in red. H:Hateful, O:Offensive, N:Normal**

|        | English | |        | Chinese | |
|--------|---------|--------|--------|---------|--------|
| **H**  | **O**   | **N**  | **H**  | **O**   | **N**  |
| like   | like    | like   | 娘炮   | 娘炮    | 一个   |
| that   | whore   | gay    | 英文   | 公主    | 他妈的 |
| shorts | know    | know   | 一个   | 一个    | 基佬   |
| faggy  | that    | shorts | 仙女   | 鸡脚    | 阴茎   |
| oh     | shorts  | that   | 日本   | 露出    | 公主   |
| know   | man     | oh     | 有没有 | 泼妇    | 花痴   |
| yeah   | going   | yeah   | 19     | 婊子    | 男同学 |
| going  | oh      | going  | covid  | 老鸨    | 男人   |
| get    | pussy   | get    | 真的   | 贱人    | 卖淫   |
| okay   | faggy   | right  | 男人   | 小黑子  | 男子   |

Women were the most frequently targeted group across both languages and categories, followed by men and LGBTQ individuals. A distinction emerges in the targeting of "*Other*" victims: English videos more frequently target individuals based on religion or race, while Chinese videos tend to focus on nationality.

## 3.4 Modality Analysis

The contribution of different modalities to the hatefulness or offensiveness of content is summarized in Table 5. Key findings include a higher prevalence of text-only hate content in English offensive videos and a significant portion of videos in both languages being classified as hateful or offensive due to multiple modalities. This underscores the importance of a multimodal approach in effectively identifying and classifying hate speech content.

**Text Analysis**: We first combine video titles and transcripts into a unified text feature, with stop words removed for clarity. For Chinese text, the Jieba Python library[3] was employed for sentence segmentation. A term frequency-inverse document frequency (tf-idf) analysis yielded insights into the most prevalent words across categories, detailed in Table 6. Notably, Chinese hateful/offensive and normal videos were found to contain a significant number of hate lexicons, suggesting that text alone, while informative, may not suffice for comprehensive hate speech detection.

**Audio Analysis**: Our evaluation of audio content focused on amplitude and Zero Crossing Rate (ZCR) metrics. Amplitude analysis of English videos, represented in Figure 1, highlighted that *Hateful* and *Offensive* typically exhibit higher sound intensities. Similarly, ZCR analysis of English videos, shown in Figure 2, indicated that these videos also feature higher rates of zero crossing,

[3]https://pypi.org/project/jieba/

2024-04-13 10:04. Page 4 of 1–9.

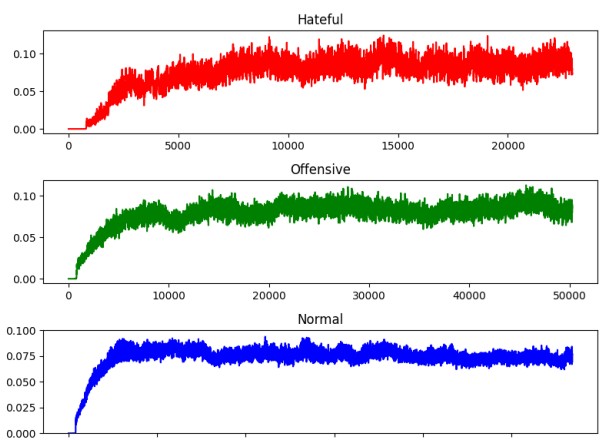

Figure 1: Amplitude of English YouTube videos.

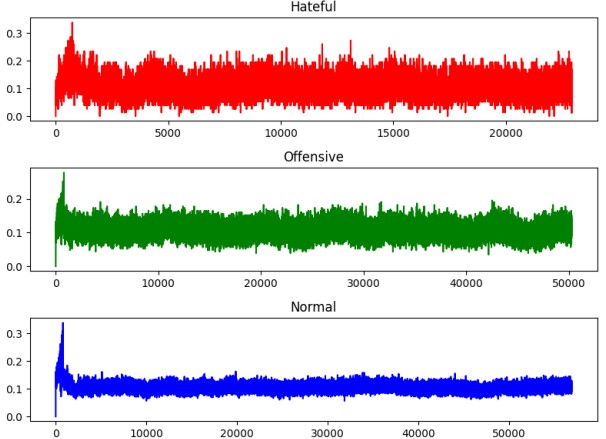

Figure 2: Zero Crossing Rate of English YouTube videos.

pointing to a potentially noisier audio profile. These patterns were consistent across languages, underscoring the audio modality's role in discerning video content quality and nature.

**Vision Analysis**: In our visual data analysis, we extracted one frame per second from each video, applying YOLOv3 for object detection on these frames [29]. This analysis revealed a notable trend: English videos categorized as Hateful and Offensive had a higher detection rate of 'person' objects (67% and 72%, respectively) compared to Normal videos (62%). Conversely, the application of YOLOv3 on Chinese videos resulted in a significant number of detection failures—717 instances for Chinese videos, compared to 277 for English. This discrepancy highlights a crucial limitation of current object detection models, which may stem from the underrepresentation of diverse gender and racial characteristics in training datasets like Coco[25, 44]. Such disparities pose substantial challenges in accurately recognizing individuals in videos from non-Western contexts, thereby impacting the reliability of our visual analysis results for Chinese content.

# 4 BENCHMARKING MODELS

In this section, we outline the approach for detecting hatefulness in videos using the MultiHateClip dataset and describe the benchmarking of various models tailored for this purpose.

## 4.1 Problem Definition

The task is to classify a given video $V$ into one of three categories: Hateful ($y = 0$), Offensive ($y = 1$), or Normal ($y = 2$). This classification considers three modalities within the video:

- **Text** ($T$): Comprising words $\{w_1, w_2, \ldots, w_m\}$, aggregated from the video's title and transcript.
- **Audio** ($A$): The auditory component of the video.
- **Vision** ($V$): A sequence of frames $\{f_1, f_2, \ldots, f_n\}$ extracted from the video.

The challenge for the model $f$ is to accurately map these modalities to the ground-truth label $y$, where $f : f(T, A, V) \rightarrow y$ and $y$ belongs to the set $\{0, 1, 2\}$. While initially treated as a multiclass classification problem, we also consider a binary classification scenario by combining Hateful and Offensive categories into a single label for a simplified offensive label analysis.

## 4.2 Preprocessing

For text data, preprocessing involves removing stop words to refine the dataset. In the case of vision-based analysis, we employ a strategy of selecting one frame per second from each video. For consistency and to accommodate videos shorter than 60 seconds, we use padding images to ensure each video analysis is based on a uniform duration of up to 60 frames.

## 4.3 Text-Based Models

**mBERT:** We incorporate mBERT (Multilingual BERT) [14] for its established effectiveness in various text classification tasks and its exceptional multilingual capabilities, which cover 104 languages. This adaptability is crucial for analyzing the linguistic diversity present in our dataset. mBERT's proficiency in hate speech detection [27] and its utility for low-resource languages [35] further justify its selection. For our analysis, the text data is fed into a pretrained mBERT model, from which we derive the 768-dimensional hidden states representing text features. These features are then passed through two fully connected (FC) layers to classify each video into the appropriate category (Hateful, Offensive, or Normal). This configuration is referred to as Model T1 in our study.

**GPT-4-Vision-Preview without vision (GPT-4):** *GPT-4-Vision-Preview* [3], known for its advanced capabilities in vision-language tasks, is also employed in a text-only capacity to leverage its sophisticated language understanding. For this experiment, we omit vision-related features, focusing solely on the model's linguistic analysis capabilities. Following the methodology described in Table 7, we prepare four demonstration examples reflective of the class distribution in our dataset—two *Normal*, one *Offensive*, and one *Hateful*. Each example consists of the video's text content, a classification query, and the actual label. We then present new video prompts to *GPT-4* in a similar format to predict their labels, delineating this approach as our T2 model.

## 4.4 Audio-Based Models

**MFCC:** Mel Frequency Cepstral Coefficients (MFCC) are a critical feature in audio signal processing, representing the short-term power spectrum of sound. MFCC's effectiveness in audio classification tasks is well-established [5, 6, 36, 42]. For audio analysis in our study, we generate a 40-dimensional MFCC vector for each audio signal to capture its essential characteristics. These vectors are then processed through two FC layers, which classify the audio into Hateful, Offensive, or Normal categories. This model is designated as A1 in our research.

## 4.5 Vision-Based Models

**ViT:** The Vision Transformer (ViT) [15] represents a significant advance in image recognition technology, employing transformer networks to analyze visual data. Its application in identifying offensive content, such as memes [28], underscores its potential for broader image-based classification tasks. In our study, we utilize ViT to process 60 frames extracted from each video, leveraging a pre-trained model to obtain a 768-dimensional feature vector for each frame. Subsequent processing of these vectors through an LSTM [22] network and two FC layers enables the classification of each video into Hateful, Offensive, or Normal categories. We designate this approach as Model V1.

## 4.6 Vision-Language Models

**GPT-4-Vision-Preview (GPT-4V):** Specifications for the GPT-4-Vision-Preview model prompts are outlined in Table 7. This model extends the capabilities of *GPT-4-Vision-Preview (w/o vision)* by incorporating visual inputs. In our setup, visual information is initially represented through textual descriptions generated by GPT-4, based on the prompt "*Please briefly summarize the video's content given the images*". We enhance visual data representation by uniformly sampling N frames from each video, combining four frames into a single image to improve information density and GPT-4's interpretative accuracy. Both demonstration examples and actual video prompts are processed by GPT-4V to predict the classification label, designating this approach as Model VL1.

**VLM:** The VLM (Vision-Language Model) [41] adopts a versatile, task-agnostic strategy for multimodal pre-training, adept at handling video, text, or combined inputs for various tasks. Outperforming other models in vision-language integration, VLM shines in video captioning and retrieval tasks [30]. In our application, VLM extracts and merges features from text and visual inputs into a 768 × 2-dimensional representation. This composite representation is then processed through two FC layers for final video categorization, with this method marked as Model VL2.

## 4.7 Multimodal Models

**VLM ⊙ MFCC:** This model, as depicted in Figure 3, utilizes a fusion strategy where features extracted by the VLM and MFCC are first processed through separate FC layers. The resultant feature vectors are then concatenated and further processed through an FC layer to classify the videos. We designate this integrated approach as Model M1.

**HateMM:** Inspired by the fusion strategy outlined in the HateMM study [11], our model, named M2, integrates outputs from mBERT,

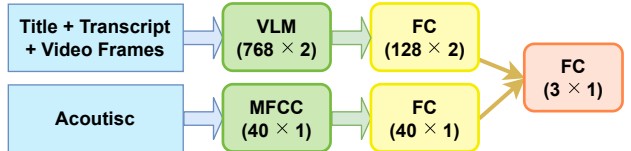

**Figure 3: Framework of the multi-modal model employing VLM and MFCC. FC: Fully Connected Layer.**

ViT, and MFCC. This model employs three FC layers to individually process text and audio features, while visual features extracted by ViT are first passed through an LSTM layer followed by an FC layer for temporal analysis. The processed modalities are subsequently concatenated and directed through an FC layer for final classification, embodying a comprehensive strategy to capitalize on the strengths of each modality.

## 5 EXPERIMENT

### 5.1 Experiments setup

We partitioned our dataset into 80% for training and 20% for testing, while maintaining identical label distributions within each subset as in the overall dataset. The configuration for each model's layers is detailed below to clarify their processing pipeline:

- **mBERT:** The 768-dimensional output from mBERT is processed through two fully connected (FC) layers with dimensions of 128 and 3, respectively.
- **MFCC:** The 40-dimensional MFCC outputs are input into two FC layers of sizes 40 and 3, respectively.
- **ViT:** ViT employs a 128-unit LSTM layer followed by two FC layers, sized 128 and 3 units.
- **VLM:** The 1536-dimensional (768x2) output from VLM is directed through two FC layers of 256 (128x2) and 3 units. Notably, VLM was not applied to Chinese videos due to its training limitations on Chinese datasets.
- **GPT-4V:** For the GPT-4 prompt configuration, we experimented with a range of 2 to 16 images (N), finding that N=4 offered the best balance between accuracy and processing efficiency for our video duration.

For models involving *VLM ⊙ MFCC* fusion, the outputs are initially processed through separate FC layers (256 and 40 units) before concatenation and final classification through a 3-unit FC layer. The *HateMM* model processes *mBERT ⊙ MFCC* outputs through two 128-unit FC layers and a 64-unit FC layer and *ViT* outputs through a 128-unit LSTM and a 64-unit FC layer, with all outputs concatenated for final classification through a 3-unit FC layer.

The output from each model's final layer is a 3-dimensional vector, which is then passed through a log-softmax layer and subjected to a cross entropy loss function for estimating video label probabilities. Training spans 40 epochs utilizing the Adam optimizer, with a batch size of 16 and a learning rate of 1e-4.

### 5.2 Evaluation metrics

To align with the conventions of existing research and to adequately address potential dataset imbalances, we employ a set of standard evaluation metrics: Macro F1 score, F1, precision, and recall. These metrics evaluate our models' performance in the classification task,

**Table 7: Example GPT-4V's prompt for *Hateful Video Classification* (reformatted for visualisation purposes). The prompt input consists of the extracted textual features (in purple), extracted vision features(in pink), the question (in green), and the GPT-4V's generated answers (in orange).**

---

\# Demonstration Examples 1...4

**Input**: Given a video with Title: "POV: GTA Players when they see a hooker", Transcript: "Do they always do that? Shit? Oh. Do we have here?..." and Vision Content: "The video appears to be a humorous and exaggerated depiction of a player's reaction..."

If the video is normal, output 'Normal'; if it is offensive, output 'Offensive'; if it is hateful, output 'Hateful'.

**Output**: Hateful

...

\# Actual Prompt

**Input**: Given a video with Title: "Mistress", Transcript: "What's the difference between a wife and a mistress about 50 pounds? later Wait, are you talking about a weight?" and Vision Content as shown in images.

Image 1                           Image 2

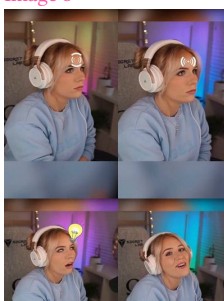 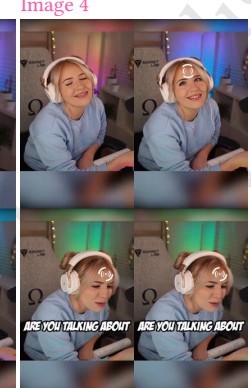

Image 3                           Image 4

If the video is normal, output 'Normal'; if it is offensive, output 'Offensive'; if it is hateful, output 'Hateful'.

**Output**:

---

distinguishing between *Hateful*, *Offensive*, and *Normal* video content. For a more streamlined analysis, we also consolidate the *Hateful* and *Offensive* categories into a single *Offensive* label, enabling us to report on binary classification outcomes as well. In our results, the highest-performing metric is denoted in **bold**, with the second-best performance underscored.

## 5.3 Performance Evaluation

The performance of various models on multiclass classification tasks is detailed on the left side of Tables 8 for English videos and 9 for Chinese videos. For English, models T1 and M2 were the top performers, whereas for Chinese, T2 and VL2 stood out. Interestingly, the addition of modalities did not translate to enhanced performance across all models. For instance, comparing VL2 to T2 and M1 to VL1 in English showed F1 score improvements of 12.8% and 2%, respectively, by incorporating additional modalities. However, such improvements were not consistent across all model comparisons.

A closer look at the F1 scores for individual labels indicated instances where the Hateful category received a score of zero, suggesting challenges in distinguishing between Hateful and Offensive content, compounded by the low representation of Hateful videos in the dataset. This scenario underscores the importance of binary classification results for a fairer comparison of model efficacy.

Binary classification results, presented in Tables 8 for English and 9 for Chinese videos, exhibited more consistent model performances. Specifically, VL2 for English and VL1 for Chinese showcased superior results among the models tested. The effectiveness of integrating multiple modalities was particularly evident in the Chinese dataset, where multimodal models outperformed their unimodal counterparts. This trend was less pronounced in English, possibly due to the reduced presence of multimodal hate speech, as per our annotation insights. Despite higher performance metrics for English videos, the top scores remain modest, highlighting the challenging nature of the MultiHateClip dataset.

## 5.4 Limitations of Existing Models

Our evaluation of the baseline models uncovers three significant challenges impacting their efficacy in hateful video detection:

**Distinguishing Between Hateful and Offensive Content:** Models struggle with discerning between hateful and offensive content, leading to frequent misclassifications. For example, the VL2 model, when applied to Chinese videos, incorrectly labeled 16 out of 22 videos identified as hateful as offensive, thereby compromising its performance in the multiclass classification compared to T2.

**Training Deficiencies on Non-Western Cultural Data:** The observed discrepancy in performance between Chinese and English videos under the same model configurations is largely attributable to the models' training on datasets dominated by Western cultural contexts. This issue is evident across text, vision, and audio models, negatively impacting their applicability to Chinese video content.

**Late Fusion of Multimodal Vectors:** Despite evidence from human annotation that multimodal content is prevalent in hateful videos, efforts to improve detection accuracy through multimodal vector fusion have seen limited success. The M2 model, despite achieving a 0.790 F1 score on the HateMM dataset for binary classification as reported by [11], only attained a 0.561 score on our English dataset. This performance did not surpass that of the unimodal T1 model and was inferior to VL1, indicating that the strategy of combining pre-trained models for different modalities with late fusion might be inadequate for handling complex multimodal data. In contrast, the success of the VL1 and VL2 models highlights the advantage of integrating visual and textual information early in the processing pipeline.

**Table 8: Model performance for English YouTube hateful video classification. H:hateful, O:offensive, M-F1:macro-F1, R:recall, P:precision.**

| | Model | Multiclass | | | | | | | Binary | | | |
|---|---|---|---|---|---|---|---|---|---|---|---|---|
| | | M-F1 | F1(H) | R(H) | P(H) | F1(O) | R(O) | P(O) | M-F1 | F1(O) | R(O) | P(O) |
| T1 | *mBERT* | **0.439** | **0.286** | **0.25** | 0.333 | 0.33 | 0.391 | 0.286 | 0.561 | 0.423 | 0.468 | 0.387 |
| T2 | *GPT-4* | 0.259 | 0.138 | 0.125 | 0.154 | 0.356 | 0.804 | 0.228 | 0.394 | 0.506 | **0.968** | 0.343 |
| A1 | *MFCC* | 0.366 | 0.229 | **0.25** | 0.211 | 0.4 | 0.717 | 0.277 | 0.47 | 0.47 | 0.758 | 0.341 |
| V1 | *ViT* | 0.36 | 0.0 | 0.0 | 0.0 | 0.278 | 0.217 | 0.385 | 0.548 | 0.295 | 0.21 | 0.5 |
| VL1 | *VLM* | 0.405 | 0.0 | 0.0 | 0.0 | 0.41 | 0.348 | 0.5 | **0.583** | 0.362 | 0.274 | **0.531** |
| VL2 | *GPT-4V* | 0.387 | 0.2 | 0.125 | **0.5** | **0.429** | **0.826** | 0.29 | 0.525 | **0.518** | 0.823 | 0.378 |
| M1 | *VLM ⊙ MFCC* | 0.425 | 0.25 | **0.25** | 0.25 | 0.282 | 0.261 | 0.308 | 0.559 | 0.376 | 0.355 | 0.4 |
| M2 | *HateMM* | **0.439** | **0.286** | **0.25** | 0.333 | 0.33 | 0.391 | 0.286 | 0.561 | 0.423 | 0.468 | 0.387 |

**Table 9: Model performance for Chinese Bilibili hateful video classification. H:hateful, O:offensive, M-F1:macro-F1, R:recall, P:precision.**

| | Model | Multiclass | | | | | | | Binary | | | |
|---|---|---|---|---|---|---|---|---|---|---|---|---|
| | | M-F1 | F1(H) | R(H) | P(H) | F1(O) | R(O) | P(O) | M-F1 | F1(O) | R(O) | P(O) |
| T1 | *mBERT* | 0.309 | 0.087 | 0.045 | **1.0** | 0.0 | 0.0 | 0.0 | 0.438 | 0.035 | 0.018 | **1.0** |
| T2 | *GPT-4* | **0.346** | **0.25** | **0.182** | 0.4 | 0.316 | **0.794** | 0.197 | 0.478 | 0.483 | **0.875** | 0.333 |
| A1 | *MFCC* | 0.279 | 0.0 | 0.0 | 0.0 | 0.0 | 0.0 | 0.0 | 0.419 | 0.0 | 0.0 | 0.0 |
| V1 | *ViT* | 0.326 | 0.0 | 0.0 | 0.0 | 0.157 | 0.118 | **0.235** | 0.506 | 0.192 | 0.125 | 0.412 |
| VL1 | *VLM* | - | - | - | - | - | - | - | - | - | - | - |
| VL2 | *GPT-4V* | 0.33 | 0.0 | 0.0 | 0.0 | **0.343** | 0.706 | 0.226 | **0.567** | **0.488** | 0.714 | 0.37 |
| M1 | *VLM ⊙ MFCC* | - | - | - | - | - | - | - | - | - | - | - |
| M2 | *HateMM* | 0.294 | 0.0 | 0.0 | 0.0 | 0.234 | 0.382 | 0.169 | 0.485 | 0.321 | 0.393 | 0.272 |

**Table 10: Video examples with Text, Visual Description, Ground Truth, and Predictions. GD:ground truth, H:hateful, O:offensive, N:normal**

| Text | Visual Description | GD | T1 | V1 | VL1 | M2 |
|---|---|---|---|---|---|---|
| MMN VS BHV(such a whore) round 1 fight round 2 final | The video shows footballers from MNN and BHV teams, highlighting MNN's superiority through a one-to-one comparison of attributes like pace, shooting, and dribbling. | N | O | O | N | O |
| Faceraping Manwhores for | Two men contorted their faces and sang in front of the camera. One playfully stuck out his tongue before the camera abruptly shifted focus to another man, ending the video. | O | N | O | O | N |
| Secret Of Big Dick | A man beside a woman catches her eye; she glances at his bottom, then asks for his contact information with delight. Suddenly, he retrieves a bottle of drinks from his pants. | H | N | N | O | N |

These findings emphasize the necessity for tailored approaches in model training and modality fusion to effectively address the nuanced and culturally diverse nature of hate speech in videos.

## 5.5 Error Analysis

Table 10 presents three illustrative examples from the English video dataset, demonstrating the importance of multimodal information in accurate content classification. The first example shows that while the language used could be perceived as hateful, the integration of corresponding visual cues clarifies that the video is actually comparing football teams and, thus not intended to be harmful. In the second example, textual data alone might be insufficient to deem the content as hateful or offensive; however, the addition of visual information clearly indicates its offensive nature. A comparison between models VL1 and M2, which both utilize textual and visual modalities, shows that VL1's early fusion approach accurately categorizes both examples, unlike M2 which fails to do so. The third example underscores a common challenge: all models, including the top-performing VL1, erroneously classify the video

as offensive rather than hateful, highlighting the ongoing difficulty in distinguishing between these two categories of content.

## 6 CONCLUSION

This study introduces MultiHateClip, a multilingual dataset for hateful video detection, enriched with fine-grained labels across English and Chinese languages. Our investigation reveals the significant advantage of vision-language models (*VLM* and *GPT-4V*) over traditional multimodal approaches employing late fusion (*HateMM* and *VLM ⊙ MFCC*). This underscores the essential role of modality integration and suggests that the early fusion of visual and textual information might be key to enhancing detection performance. The findings suggests that the exploration of sophisticated modality fusion techniques, setting a new direction for future research in the nuanced domain of hate speech detection. MultiHateClip not only contributes a valuable resource for advancing this field but also highlights the critical need for multimodal analysis in understanding and combating hate speech in a multilingual context.

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
