# OpenReview forum: "MultiHateClip: A Multilingual Benchmark Dataset for Hateful Video Detection on YouTube and Bilibili"
_acmmm.org/ACMMM/2024/Conference — MM2024 Oral_

### Official Review · Reviewer_3Twx · 2024-05-01

**Rating:** 2
**Confidence:** 2

**Summary:**

This paper presents a novel dataset on hateful video detection. This dataset is composed of both English videos collected from Youtube and Chinese videos collected from Bilibili. Different from prior work, the dataset contains three labels: Hateful, Offensive and Normal, with supplemental information, e.g., hateful segment. This paper gives a detailed description on the annotation process and evaluates existing models on the collected dataset. In parallel, the author characterizes the dataset and analyzes the model performances.

**Strengths:**

1. This paper has a clear writing structure - it’s easy to read and follow.
2. This paper focuses on a meaningful research topic - hateful video detection, and is the first to collect a multilingual multiclass dataset on it.
3. This paper carries out a comprehensive analysis on the dataset and evaluation on the existing models. Moreover, it gives insights through the result analysis.

**Limitations:**

1. In 3.1 Q1), the definitions of Hateful and Offensive seem overlapping. It’s not very clear what distinguishes Offensive from Hateful. In 3.2 Q2), the definitions of the start and end times of the hateful segment is unclear. For example, [3s, 5s] could be a hateful segment, whilst a larger time window, [2s, 6s], could also be true.
2. Figure 1 and 2 do not specify the meanings of x-axis and y-axis. Moreover, Y-axis should be aligned across the subfigures, to have a clear and fair comparison among these three labels. Also, it seems to me that Offensive and Normal’s Amplitude and Zero Crossing Rate graph are similar? It looks different in the figures only because the y-axis are not aligned?
3. In 4, the paper doesn’t include models proposed for audio/video classification. Rather, it only includes general-purposed large language and/or vision models. It might be better to include some SOTA video classification models as well.
4. In 6, it says that “our investigation reveals the significant advantage of VLM and GPT-4V over HateMM and VLMxMFCC”. However, in Table 8, the latter outperforms the former in many metrics. It seems contradictory that late fusion is disadvantaged?
5. Why is that in Table 8 the first row and the last row are having the exactly same numbers?
6. In 5.5, the error analysis on only three examples sounds a little bit less convincing. Could you include more examples and examine some group characteristics?
7. It’s better for the numbers to have the same decimal places in the tables.

(Minor) 8. The writing needs to be polished in this paper - there are some less commonly-used expressions, e.g., l.510 “crucial limitation” and l.82 “Videos harness the synergistic potential”. Although they are understandable, it makes the reading experience less smooth. Furthermore, this paper uses a lot of past tenses, where the present tense is supposed to be used instead.

**Suitability:**

3

---

### Official Review · Reviewer_3UNE · 2024-05-20

**Rating:** 5
**Confidence:** 4

**Summary:**

This study introduces MultiHateClip, a novel multilingual dataset curated using hate lexicons and human annotation. It aims to improve hateful video detection on platforms like YouTube and Bilibili, covering content in English and Chinese. With 2,000 annotated videos, it offers a cross-cultural perspective on gender-based hate speech. Evaluations of state-of-the-art models underscore the need for multimodal and culturally nuanced approaches in combating online hate speech.

**Strengths:**

1.	Hateful Video Detection is a problem worthy of attention. From what I understand, data resources in this area are extremely scarce. This paper considers both Chinese and English languages, which is helpful for researchers in related fields to carry out their work.
2.	MultiHateClip provides a fine-grained annotation, and the scale of the data is comparable to existing resources and even larger.
3.	Researchers introduced multiple baselines to make full evaluations on the dataset for reference. As a resource paper, the experimental quantity is sufficient.

**Limitations:**

1.	It is recommended to provide more details on dataset construction, including the time period of data sampling and demographic information of annotators, such as race, age, etc.
2.	In Section 3.3, the researchers did not further discuss based on the consistency scores, and samples with annotation discrepancies may also pose challenges in detection.
3.	For improved readability, the data in Table 9 and Table 10 could be formatted consistently, for example, retaining three decimal places.
4.	Considering that the dataset proposed in this paper contains Chinese samples, it is necessary to cite related papers about Chinese hate speech detection to explain the characteristics of Chinese culture. The following are some representative works:
[1] COLD: A Benchmark for Chinese Offensive Language Detection. EMNLP2023.
[2] Facilitating fine-grained detection of Chinese toxic language: Hierarchical taxonomy, resources, and benchmarks. ACL2023.
[3] SWSR: A Chinese dataset and lexicon for online sexism detection. Online Social Networks and Media.

**Suitability:**

3

---

### Official Review · Reviewer_KPrU · 2024-05-20

**Rating:** 6
**Confidence:** 3

**Summary:**

This paper presents MultiHateClip, a novel dataset comprising 2,002 code-mixed (English and Chinese) videos obtained from multiple platforms annotated across four aspects. It also presents a comparison of the performance of various SoTA video classification models and provides insights on the drawbacks of each. The paper also highlights the importance of multimodal analysis from a cultural standpoint.

**Strengths:**

1. Tackling the problem of Western contextualization and bias in most video datasets is realistic and novel.

2. The dataset is annotated across four aspects, and previous datasets that annotate for a single aspect (usually hate) leave out a lot of important context and information. They also include annotations for audio, video, and text. This makes the dataset more comprehensive and makes more data available for future users of the dataset.

3. The dataset obtains data from two video platforms, which may reduce any inherent biases caused by the particular platform's guidelines, and the videos they may promote or suppress. The dataset size is also larger than previous datasets in this domain.

4. The annotation process is sound and well-described. The inter-annotator agreement is increased after making changes to the annotation system.

5. The performance of models with a varied range of sizes (from mBERT to GPT-4V) is compared with insights into each model's performance.

6. Each part of the paper remains close to the problem of video contextualization and cultural dynamics, making it cohesive overall.

**Limitations:**

1. Although the dataset size is greater than previous datasets (e.g. HateMM), it still might not be enough to train large multimodal models, LLMs, etc. that require a huge amount of data.

2. Since this dataset aims to provide ML models with more context for generalization, could this dataset be expanded to be annotated across even more aspects? Although this annotation process is comprehensive, it is hard to say if it completely covers the message of an entire video.

3. While readers who work in this domain are familiar with the Western bias in previous datasets, most readers might not be aware of this. Highlighting this bias with empirical data (besides attributing it to the unknown training data of pre-trained models) would strengthen this paper's argument significantly.

4. Despite paper length limitations, the performance of the compared models could be discussed further (e.g. why does T1 perform better than VL models in Table 8?). Shortening sections 4.3 - 4.7 may be considered as it introduces well-known architectures.

However, these weaknesses are minor or only regarding the paper's presentation and are outweighed by the strengths of this dataset.

**Suitability:**

3

---

### Meta-Review · Area_Chair_gMiN · 2024-07-02

**Recommendation:** Accept (Oral)
**Confidence:** 4

**Metareview:**

This paper provides a new multilingual benchmark dataset for hateful video detection (especially focusing on YouTube and Bilibili platforms). All reviewers think that the addressed issue is important and realistic and the annotated dataset is a significant contribution to the research community in this area. Though this paper received a negative recommendation (with a score list of 3, 4, 5), I feel the remained issues are not hard to tackle when preparing the camera-ready version. Overall, I recommend the acceptance of this submission, but the authors are still encouraged to revise the manuscript according to the detailed reviews.